# A Technical Critique of Some Parts of the Free Energy Principle

**DOI:** 10.3390/e23030293

**Published:** 2021-02-27

**Authors:** Martin Biehl, Felix A. Pollock, Ryota Kanai

**Affiliations:** 1Araya Inc., Tokyo 107-6024, Japan; kanair@araya.org; 2School of Physics and Astronomy, Monash University, Clayton, VIC 3800, Australia

**Keywords:** free energy principle, stochastic differential equations, Markov blanket

## Abstract

We summarize the original formulation of the free energy principle and highlight some technical issues. We discuss how these issues affect related results involving generalised coordinates and, where appropriate, mention consequences for and reveal, up to now unacknowledged, differences from newer formulations of the free energy principle. In particular, we reveal that various definitions of the “Markov blanket” proposed in different works are not equivalent. We show that crucial steps in the free energy argument, which involve rewriting the equations of motion of systems with Markov blankets, are not generally correct without additional (previously unstated) assumptions. We prove by counterexamples that the original free energy lemma, when taken at face value, is wrong. We show further that this free energy lemma, when it does hold, implies the equality of variational density and ergodic conditional density. The interpretation in terms of Bayesian inference hinges on this point, and we hence conclude that it is not sufficiently justified. Additionally, we highlight that the variational densities presented in newer formulations of the free energy principle and lemma are parametrised by different variables than in older works, leading to a substantially different interpretation of the theory. Note that we only highlight some specific problems in the discussed publications. These problems do not rule out conclusively that the general ideas behind the free energy principle are worth pursuing.

## 1. Overview

In [1], it was argued that the internal coordinates of an ergodic random dynamical system with a Markov blanket necessarily appear to engage in active Bayesian inference. Here, we reproduce the argument supporting this interpretation in detail and highlight at which points it faces technical issues. In the course of our critique, we also mention issues of some closely related alternative arguments. In cases where our results have clear consequences for the more recent related publications [2,3], we also mention those. In particular, we point out a conceptual difference in these latter works that has not previously been acknowledged. However, our analysis thereof does not go beyond a few remarks. In an additional section, we discuss the effect of our argument on [4]. The logical structure of the present paper is depicted in Figure 1. We note that the technical issues presented here do not affect the validity of approaches where a (expected) free energy minimizing agent is assumed a priori, as presented in, e.g., [5]. None of [1,2,3,4] make this assumption; they instead aim to identify the conditions under which such agents will emerge within a given stochastic process. We criticize specific formal issues in the latter publications but leave open whether they can be fixed. We now briefly introduce the setting of [1] and then sketch the content of this paper. We now briefly introduce the setting of [1] and then sketch the content of this paper.

The starting point is a random dynamical system whose evolution is governed by the stochastic differential equation:(1)$$\begin{array}{c} \dot{x}=f\left(x\right)+\omega , \end{array}$$
where the system state *x* and vector field f(x) are multi-dimensional and ω is a Gaussian noise term. There is an additional assumption that the system is ergodic, such that the steady state probability density $p^{*}\left(x\right)$ is well defined (In the original paper, the ergodic density is simply denoted p(x). We here add a star to highlight that it is a time independent probability density.). In this case, $-\ln p^{*}\left(x\right)$ plays the role of a potential function, in the sense that *f* can be formulated in terms of its gradients [6,7].

It is then assumed that there is a coordinate system x=(ψ,s,a,λ) with $\psi =(\psi_{1},...,\psi_{n_{\psi }})$, $s=(s_{1},...,s_{n_{s}})$, $a=(a_{1},...,a_{n_{a}})$, and $\lambda =(\lambda_{1},...,\lambda_{n_{\lambda }})$, referred to as external, sensory, active, and internal coordinates (these are called “states” in [1]), respectively, such that the following condition holds:

**Condition** **1.**
*The function f(x) can be written as:*
(2)$$\begin{array}{c} \begin{array}{cc} f(x) & =\left(\begin{array}{c} f_{\psi }\left(\psi ,s,a\right)\\ f_{s}\left(\psi ,s,a\right)\\ f_{a}\left(s,a,\lambda \right)\\ f_{\lambda }\left(s,a,\lambda \right) \end{array}\right). \end{array} \end{array}$$


This particular structure is described as “[formalizing] the dependencies implied by the Markov blanket” [1]. In contrast, more recent works [2,3] formulated the Markov blanket in terms of the statistical dependencies of the ergodic density $p^{*}\left(x\right)=p^{*}\left(\psi ,s,a,\lambda \right)$. Specifically, the following condition is presented:

**Condition** **2.**
*The ergodic density factorises as:*
(3)$$\begin{array}{c} p^{*}\left(\psi ,s,a,\lambda \right)=p^{*}\left(\psi |s,a\right)p^{*}\left(\lambda |s,a\right)p^{*}\left(s,a\right). \end{array}$$


In other words, the internal and external coordinates are independently distributed when conditioned on the sensory and active coordinates. This means we have two different formal expressions of what constitutes a Markov blanket in these publications, and their relationship has not previously been established.

Taking Condition 1 to hold, the argument of [1] then proceeds along the following steps:**Step** **2**Rewrite the vector field f(ψ,s,a,λ) describing the dynamics of the system in terms of the gradient of the negative logarithm of the ergodic density $p^{*}\left(\psi ,s,a,\lambda \right)$ of that system.**Step** **3**Rewrite the components $f_{\lambda }\left(s,a,\lambda \right)$ and $f_{a}\left(s,a,\lambda \right)$ of the vector field f(ψ,s,a,λ) in terms of only partial gradients of the negative logarithm of $p^{*}\left(\psi ,s,a,\lambda \right)$.**Step** **4**Assert (in the free energy lemma) the existence of a density q(ψ|λ) over the external coordinates ψ parameterized by the internal coordinates λ and that f(ψ,s,a,λ) can again be rewritten, this time in terms of a free energy depending on q(Ψ|λ) (here, and whenever it would otherwise be ambiguous, we use a capitalized Ψ to indicate full distributions, rather than the probability density for a specific value of ψ).**Step** **5**Claim that the equivalence of the equations of motion in Step 3 and Step 4 implies that certain partial gradients of the KL divergence between q(Ψ|λ) and the conditional ergodic density $p^{*}\left(\Psi |s,a,\lambda \right)$ must vanish.**Step** **6**Claim that it follows from Step 5 that q(Ψ|λ) and $p^{*}\left(\Psi |s,a,\lambda \right)$ are “rendered” equal.**Step** **7**Interpret:$p^{*}\left(\Psi |s,a,\lambda \right)$ as a posterior over external coordinates given particular values of sensor, active, and internal coordinates,q(Ψ|λ) as encoding Bayesian beliefs about the external coordinates by the internal coordinates, andtheir equality as the internal coordinates appearing to “solve the problem of Bayesian inference”.

In the present paper, we make the following main observations:The re-expression of Equation (Equation 1) in the form chosen in Step 2 is derived under restrictive assumptions, including that the system is subject to Gaussian and Markov noise.Conditions 1 and 2 are independent of each other.Conditions 1 and 3 together lead to a system where the interpretation of *s* and *a* as sensory and active coordinates is questionable.Under both Conditions 1 and 2, the expressions of $f_{\lambda }\left(s,a,\lambda \right)$ and $f_{a}\left(s,a,\lambda \right)$ resulting from Step 3 are not as general as those contained in the result of Step 2. The more general alternative expression derived in [2] remains insufficiently general.Under both Conditions 1 and 2, the free energy lemma, when taken at face value, is wrong and cannot be salvaged by using alternatives in Step 3.Under both Conditions 1 and 2, contrary to Step 6, the vanishing of the gradient of the KL divergence does not imply the equality of q(Ψ|λ) and $p^{*}\left(\Psi |s,a,\lambda \right)$.As a consequence, the basic preconditions for the interpretations in Step 7 are not implied by either of the two proposed Markov blankets Conditions 1 and 2.

The latter [4] presents an argument almost identical to the one in the original [1]. In Section 8, we discuss how our observations apply to this publication.

## 2. Expression via the Gradient of the Ergodic Density

Here, we introduce the expression of the system’s dynamics Equation (Equation 1) in the form used for the free energy lemma (Lemma 2.1 in [1]). This form expresses the dynamics of the internal and active coordinates of the given ergodic random dynamical system in terms of the gradient of the ergodic density $p^{*}\left(x\right)$. In accordance with the results of [7], f(x) is rewritten as (see Equation (2.5) in [1]):(4)$$\begin{array}{cc} f(x) & =\left(\Gamma +R\right)\cdot \nabla \ln p^{*}\left(x\right), \end{array}$$
where Γ is the diffusion matrix, which we will take to be block diagonal (in [1], and later work such as [2], Γ is taken to be proportional to the identity matrix), and *R* is an antisymmetric matrix, defined through the relation:(5)$$\begin{array}{c} MR+RM^{T}=M\Gamma -\Gamma M^{T}, \end{array}$$
with
(6)$$\begin{array}{c} M_{ij}=\nabla_{j}f_{i}\left(x\right). \end{array}$$
Here, and in all of [1,2,3,4], both Γ and *R* are assumed constant. We emphasise here that, for general nonlinear models, these matrices can vary with the coordinates and Equation (Equation 5) holds only approximately [8,9] (the exact conditions under which these matrices can be chosen to be constant can be found in [9,10] and, for the discrete state case, [11]). Moreover, Equation (Equation 4) is derived in the literature under the explicit assumption that the fluctuations ω be Gaussian and Markov [6,7]. For the counterexamples we present here, we restrict ourselves to the class of Ornstein–Uhlenbeck processes, for which *R* and Γ are always constant, and the ergodic density $p^{*}\left(x\right)=p^{*}\left(\psi ,s,a,\lambda \right)$ is necessarily a multivariate Gaussian with zero mean. Specifically, following [7],
(7)$$\begin{array}{c} p^{*}\left(\psi ,s,a,\lambda \right):=\frac{1}{Z}\exp\left[-\frac{1}{2}\left(\psi ,s,a,\lambda \right)U{\left(\psi ,s,a,\lambda \right)}^{⊤}\right], \end{array}$$
where (ψ,s,a,λ) is a row vector and *Z* is a suitable normalisation constant. From Equation (Equation 4), it can be seen that,
(8)$$\begin{array}{c} U=-{\left(\Gamma +R\right)}^{-1}M; \end{array}$$
though we emphasise here that strict relations between *M* and *U* can only be made because of the assumption that Γ and *R* are coordinate independent [12]. This concludes Step 2.

Before moving on to Step 3, we note that, under the assumptions implicit in Step 2, we can express Conditions 1 and 2 in terms of the matrices *M* and *U* (in the nonlinear case, these matrices can still be defined in terms of the derivatives of the force vector field and potential, respectively; however, they will be generally coordinate-dependent, even when Γ and *R* are not [8]). Firstly, since it effectively states that $\nabla_{\psi }f_{a}\left(x\right)=\nabla_{\psi }f_{\lambda }\left(x\right)=\nabla_{\lambda }f_{s}\left(x\right)=\nabla_{\lambda }f_{\psi }\left(x\right)=0$,
(9)$$\begin{array}{c} \mathrm{Condition}1\;\;\Leftrightarrow \;\;M_{a\psi }=M_{\lambda \psi }=M_{s\lambda }=M_{\psi \lambda }=0, \end{array}$$
with $M_{\alpha \beta }$ a block sub-matrix of *M* in general. Secondly, because of the multivariate Gaussian nature of $p^{*}\left(\psi ,s,a,\lambda \right)$, the dependencies of conditional distributions are encoded in the inverse *U* of the covariance matrix; we therefore have that:(10)$$\begin{array}{c} \mathrm{Condition}2\;\;\Leftrightarrow \;\;U_{\psi \lambda }=U_{\lambda \psi }=0, \end{array}$$
where $U_{\alpha \beta }$ is a block sub-matrix of *U*. These implications bring us to our first observation:

**Observation** **1.**
*Neither Condition 1 (the vector field dependency structure) nor Condition 2 (conditional independence in the ergodic distribution) imply the other:*
(11)$$\begin{array}{cc} \; & \mathrm{Condition}1\;\;⇏\;\;\mathrm{Condition}2 \end{array}$$
(12)$$\begin{array}{cc} \; & \mathrm{Condition}1\;\;⇍\;\;\mathrm{Condition}2. \end{array}$$


**Proof.** In Appendix A, we provide direct counterexamples, using the equivalent constraints on the matrices *M* and *U* in Equations (Equation 9) and (Equation 10), for the implication in either direction. That is, there exists a system obeying Condition 1 that does not obey Condition 2 (proving Equation (Equation 11)), and there exists one obeying Condition 2 that does not obey Condition 1 (proving Equation (12)). □

Henceforth, unless otherwise stated, we will assume both Conditions 1 and 2. Any implications that fail to hold in this special case cannot hold generally.

## 3. Re-Expression Using Only Partial Gradients

For Step 3, we focus on the components $f_{\lambda }=\left(f_{\lambda_{1}},\ldots ,f_{n_{\lambda }}\right)$ and $f_{a}=\left(f_{a_{1}},\ldots ,f_{n_{a}}\right)$ of *f*. Without loss of generality, we can rewrite them from Equation (Equation 4) as:
(13)$$\begin{array}{cc} f_{a}\left(s,a,\lambda \right)= & \left(R_{a\psi }\cdot \nabla_{\psi }+R_{as}\cdot \nabla_{s}+\left(\Gamma_{aa}+R_{aa}\right)\cdot \nabla_{a}\right.\\ & \;\left.+R_{a\lambda }\cdot \nabla_{\lambda }\right)\ln p^{*}\left(\psi ,s,a,\lambda \right), \end{array}$$
(14)$$\begin{array}{cc} f_{\lambda }\left(s,a,\lambda \right)= & \left(R_{\lambda \psi }\cdot \nabla_{\psi }+R_{\lambda s}\cdot \nabla_{s}+\left(\Gamma_{\lambda \lambda }+R_{\lambda \lambda }\right)\cdot \nabla_{\lambda }\right.\\ & \;\left.+R_{\lambda a}\cdot \nabla_{a}\right)\ln p^{*}\left(\psi ,s,a,\lambda \right), \end{array}$$
where $\Gamma_{nm}$ ($R_{nm}$) is the block of Γ (*R*) connecting derivatives with respect to the *m* coordinates to the time derivatives of the *n* coordinates. The expectation value with respect to $p^{*}\left(\psi |s,a,\lambda \right)$ leaves the left-hand side of these equations unchanged. A few manipulations ([2] cf. Equation (12.14), p. 129) reveal that, on the right-hand side, this leads to the ergodic density $p^{*}\left(\psi ,s,a,\lambda \right)$ being replaced by the marginalised ergodic density $p^{*}\left(s,a,\lambda \right)$ so that we get:
(15)$$\begin{array}{cc} f_{a}\left(s,a,\lambda \right)= & \left(R_{a\psi }\cdot \nabla_{\psi }+R_{as}\cdot \nabla_{s}+\left(\Gamma_{aa}+R_{aa}\right)\cdot \nabla_{a}\right.\\ & \;\left.+R_{a\lambda }\cdot \nabla_{\lambda }\right)\ln p^{*}\left(s,a,\lambda \right) \end{array}$$
(16)$$\begin{array}{cc} f_{\lambda }\left(s,a,\lambda \right)= & \left(R_{\lambda \psi }\cdot \nabla_{\psi }+R_{\lambda s}\cdot \nabla_{s}+\left(\Gamma_{\lambda \lambda }+R_{\lambda \lambda }\right)\cdot \nabla_{\lambda }\right.\\ & \;\left.+R_{\lambda a}\cdot \nabla_{a}\right)\ln p^{*}\left(s,a,\lambda \right). \end{array}$$
Since $\nabla_{\psi }\ln p^{*}\left(s,a,\lambda \right)=0$, the terms involving $\nabla_{\psi }$ drop out: (17)$$\begin{array}{cc} f_{a}\left(s,a,\lambda \right) & =\left(R_{as}\cdot \nabla_{s}+\left(\Gamma_{aa}+R_{aa}\right)\cdot \nabla_{a}+R_{a\lambda }\cdot \nabla_{\lambda }\right)\ln p^{*}\left(s,a,\lambda \right), \end{array}$$(18)$$\begin{array}{cc} f_{\lambda }\left(s,a,\lambda \right) & =\left(R_{\lambda s}\cdot \nabla_{s}+R_{\lambda a}\cdot \nabla_{a}+\left(\Gamma_{\lambda \lambda }+R_{\lambda \lambda }\right)\cdot \nabla_{\lambda }\right)\ln p^{*}\left(s,a,\lambda \right). \end{array}$$
We are not aware of how to further simplify this equation without additional assumptions. However, in (Equations (2.5) and (2.6) of [1]), all of the off-diagonal terms are implicitly assumed to vanish, i.e., Equation (Equation 4) is equated with: (19)$$\begin{array}{cc} f_{a}\left(s,a,\lambda \right) & =\left(\Gamma_{aa}+R_{aa}\right)\cdot \nabla_{a}\ln p^{*}\left(s,a,\lambda \right), \end{array}$$(20)$$\begin{array}{cc} f_{\lambda }\left(s,a,\lambda \right) & =\left(\Gamma_{\lambda \lambda }+R_{\lambda \lambda }\right)\cdot \nabla_{\lambda }\ln p^{*}\left(s,a,\lambda \right). \end{array}$$
This equation is the result of Step 3.

More recently (Appendix B of [2]), a more detailed discussion of Equation (Equation 4) was presented, where it was claimed that Condition 1 implies Condition 2 (cf. our Observation 1) along with the following simplification of Equations (Equation 17) and (18) ([2], Equations (12.8)–(12.11), (12.15), pp. 126–129): (21)$$\begin{array}{cc} f_{a}\left(s,a,\lambda \right) & =\;\left(\left(\Gamma_{aa}+R_{aa}\right)\cdot \nabla_{a}+R_{a\lambda }\cdot \nabla_{\lambda }\right)\ln p^{*}\left(s,a,\lambda \right), \end{array}$$(22)$$\begin{array}{cc} f_{\lambda }\left(s,a,\lambda \right) & =\;\left(R_{\lambda a}\cdot \nabla_{a}+\left(\Gamma_{\lambda \lambda }+R_{\lambda \lambda }\right)\cdot \nabla_{\lambda }\right)\ln p^{*}\left(s,a,\lambda \right). \end{array}$$

However, Equations (Equation 21) and (22) are still provably less general than Equations (Equation 13) and (Equation 14), even when both Conditions 1 and 2 are satisfied.

**Observation** **2.**
*Given a random dynamical system obeying Equation (Equation 1), ergodicity, and both Conditions 1 and 2, none of Equations (Equation 19)–(22) generally hold.*


**Proof.** By counterexample, see Appendix B. There, we show explicitly that a model satisfying the above assumptions does not satisfy the equations in question. □

In order to arrive at Equations (Equation 21) and (22) from Equations (Equation 17) and (18) in general, one must remove the offending “solenoidal flow” terms by fiat. That is, one assumes $R_{as}=R_{\lambda s}=0$. In [2], Equation (12.4), the following, even stronger, condition was assumed as an alternative starting point (along with Condition 2):

**Condition** **3.**
*The blocks of the R matrix appearing in Equation (Equation 4) coupling (s,a) coordinates to λ and ψ coordinates and ψ coordinates to λ coordinates vanish, i.e.,*
(23)$$\begin{array}{c} R_{\psi s}=R_{\psi a}=R_{\psi \lambda }=R_{s\lambda }=R_{a\lambda }=0. \end{array}$$


This is claimed to imply $M_{\psi \lambda }=M_{\lambda \psi }=0$, but not the full Condition 1. However, in [3], both Conditions 1 and 3 were assumed (along with $R_{as}=0$). This prompts our next observation.

**Observation** **3.**
*In a system satisfying both Conditions 1 and 3, the internal coordinates cannot be directly influenced by the sensory coordinates: $f_{\lambda }\left(s,a,\lambda \right)=f_{\lambda }\left(a,\lambda \right)$, and the external coordinates cannot be directly influenced by the active coordinates: $f_{\psi }\left(\psi ,s,a\right)=f_{\psi }\left(\psi ,s\right)$.*


**Proof.** From Equation (Equation 5), it follows that:
(24)$$\begin{array}{c} M=\left(\Gamma +R\right)M^{T}{\left(\Gamma -R\right)}^{-1}, \end{array}$$
with the inverse replaced by a pseudoinverse if Γ−R is not invertible. Therefore, if $\Gamma_{\alpha \beta }=\delta_{\alpha \beta }\Gamma_{\alpha \alpha }$ and $R_{\alpha \beta }=\delta_{\alpha \beta }R_{\alpha \alpha }$ for blocks of coordinates labelled by α and β, then:
(25)$$\begin{array}{c} M_{\alpha \beta }=\left(\Gamma_{\alpha \alpha }+R_{\alpha \alpha }\right)M_{\beta \alpha }^{T}{\left(\Gamma_{\beta \beta }-R_{\beta \beta }\right)}^{-1}, \end{array}$$
and $M_{\beta \alpha }=0\Rightarrow M_{\alpha \beta }=0$.Condition 3 implies that only the nonzero blocks of *R* are $R_{\psi \psi }$, $R_{ss}$, $R_{sa}$, $R_{as}$, $R_{aa}$, and $R_{\lambda \lambda }$, and Γ is assumed to be block diagonal. As noted in Equation (Equation 9), Condition 1 requires that $M_{a\psi }=M_{\lambda \psi }=M_{s\lambda }=M_{\psi \lambda }=0$. Through Equation (Equation 25), these together imply that $M_{\lambda s}=M_{\psi a}=0$, and hence that:
(26)$$\begin{array}{c} \begin{array}{cc} f(x) & =\left(\begin{array}{c} f_{\psi }\left(\psi ,s\right)\\ f_{s}\left(\psi ,s,a\right)\\ f_{a}\left(s,a,\lambda \right)\\ f_{\lambda }\left(a,\lambda \right) \end{array}\right), \end{array} \end{array}$$
as shown. □

In this case, the four sets of coordinates interact in a chain, and it is questionable whether the *s* and *a* coordinates can be meaningfully interpreted, respectively, as sensory inputs to the internal coordinates or their boundary-mediated influence on the external coordinates.

## 4. Free Energy Lemma

The relation of the dynamics of the internal coordinates to Bayesian beliefs is made by introducing a density (called the variational density) q(Ψ|λ) that is then interpreted as encoding a Bayesian belief. It is parameterized by the internal coordinates λ and claimed to be “arbitrary”. We take this “at face value” and consider q(Ψ|λ) to be parameterized only by λ and, therefore, to be independent of (s,a). (We note that there is a convention in the literature on variational Bayesian inference, e.g., in [13], to drop the observed variables/data in the variational density. It is possible that in [1], (s,a) was seen as observed variables and dropped from the variational density q(Ψ|λ) as in this convention. However, the reason that dropping the observed variables is justified in the established convention is that those observed variables are fixed throughout the minimization of the variational free energy and the parameters of the variational density do not influence the observed data in any way. In other words, the variational density is optimized for a single data point. In [1], the data point was continuously changing and partially doing so with dependence on the parameter λ as $\dot{a}=f_{a}\left(s,a,\lambda \right)$. These differences and their consequences are non-trivial and beyond the scope of this paper, so we assume that the variational density does not depend on (s,a).) If q(Ψ|λ) is allowed to depend on (s,a), Observation 4 does not apply, and the free energy lemma is made trivially true by setting $q\left(\psi |s,a,\lambda \right):=p^{*}\left(\psi |s,a,\lambda \right)$. The existence of the variational density q(Ψ|λ) is asserted by the free energy lemma (see Lemma 2.1 in [1]) (Explicitly, the free energy lemma asserts the existence of a free energy F(s,a,λ) in terms of which f(ψ,s,a,λ) can be expressed and not the existence of q(Ψ|λ). However, since the free energy is defined as a functional of q(Ψ|λ), it exists if and only if a suitable q(Ψ|λ) exists.).

More precisely, the free energy lemma (and Step 4) asserts that for every ergodic density (equivalently as expressed in [1], for every Gibbs energy $G\left(x\right):=-\ln p^{*}\left(\psi ,s,a,\lambda \right)$) $p^{*}\left(\psi ,s,a,\lambda \right)$ of a system obeying Equations (Equation 19) and (20), there is a free energy F(s,a,λ), defined as:(27)$$\begin{array}{cc} F(s,a,\lambda ): & =-\ln p^{*}\left(s,a,\lambda \right)+\int q\left(\psi |\lambda \right)\ln\frac{q(\psi |\lambda )}{p^{*}\left(\psi |s,a,\lambda \right)}\;d\;\psi \end{array}$$(28)$$\begin{array}{cc} \; & =-\ln p^{*}\left(s,a,\lambda \right)+D_{\mathrm{KL}}[q\left(\Psi |\lambda \right)||p^{*}\left(\Psi |s,a,\lambda \right)], \end{array}$$
in terms of the “posterior density” $p^{*}\left(\Psi |s,a,\lambda \right)$ (here, we keep the conditioning argument λ, as in [1], and do not explicitly assume Condition 2, though our conclusions are unaffected by it), such that Equations (Equation 19) and (20) can be rewritten as: (29)$$\begin{array}{cc} f_{a}\left(s,a,\lambda \right) & =-{\left(\Gamma +R\right)}_{aa}\cdot \nabla_{a}F\left(s,a,\lambda \right), \end{array}$$(30)$$\begin{array}{cc} f_{\lambda }\left(s,a,\lambda \right) & =-{\left(\Gamma +R\right)}_{\lambda \lambda }\cdot \nabla_{\lambda }F\left(s,a,\lambda \right). \end{array}$$

It is worth considering what a proof of the free energy lemma could look like. A proof of the existence of a free energy (and therefore of the free energy lemma) would need to show that, for every system satisfying the given assumptions, there always exists a q(Ψ|λ) such that the right-hand sides of Equations (Equation 29) and (Equation 30) are equal to the right-hand sides of Equations (Equation 19) and (Equation 20). Expanding Equations (Equation 29) and (Equation 30) using (28) leads to:(31)fa(s,a,λ)=(Γ+R)aa·∇alnp*(s,a,λ)(Γ+R)aa−(Γ+R)aa·∇aDKL[q(Ψ|λ)||p*(Ψ|s,a,λ)],(32)fλ(s,a,λ)=(Γ+R)λλ·∇λlnp*(s,a,λ)(Γ+R)λλ−(Γ+R)λλ·∇λDKL[q(Ψ|λ)||p*(Ψ|s,a,λ)].
For the equality of the right-hand sides to those of Equations (Equation 19) and (20), we need: (33)$$\begin{array}{cc} {\left(\Gamma +R\right)}_{aa}\cdot \nabla_{a}D_{\mathrm{KL}}[q\left(\Psi |\lambda \right)||p^{*}\left(\Psi |s,a,\lambda \right)] & =0 \end{array}$$(34)$$\begin{array}{cc} {\left(\Gamma +R\right)}_{\lambda \lambda }\cdot \nabla_{\lambda }D_{\mathrm{KL}}[q\left(\Psi |\lambda \right)||p^{*}\left(\Psi |s,a,\lambda \right)] & =0. \end{array}$$
In other words, these equations say that the free energy lemma holds if any of the following three conditions (of strictly increasing strengths) are given:There is a q(Ψ|λ) such that the partial gradients $\nabla_{a}$ and $\nabla_{\lambda }$ of the KL divergence between the variational density and the conditional ergodic density are elements of the nullspaces of ${\left(\Gamma +R\right)}_{aa}$ and ${\left(\Gamma +R\right)}_{\lambda \lambda }$, respectively.There is a q(Ψ|λ) such that the gradients of the KL divergence to $p^{*}\left(\Psi |s,a,\lambda \right)$ are equal to the nullvector:
(35)$$\begin{array}{c} \nabla_{a}D_{\mathrm{KL}}[q\left(\Psi |\lambda \right)||p^{*}\left(\Psi |s,a,\lambda \right)]=0, \end{array}$$
(36)$$\begin{array}{c} \nabla_{\lambda }D_{\mathrm{KL}}[q\left(\Psi |\lambda \right)||p^{*}\left(\Psi |s,a,\lambda \right)]=0, \end{array}$$Then, they are always elements of the nullspaces of ${\left(\Gamma +R\right)}_{aa}$ and ${\left(\Gamma +R\right)}_{\lambda \lambda }$, respectively.There is a q(Ψ|λ) such that $q\left(\Psi |\lambda \right)=p^{*}\left(\Psi |s,a,\lambda \right)$ (and hence, $p^{*}\left(\Psi |s,a,\lambda \right)=p^{*}\left(\Psi |\lambda \right)$), which implies that the KL divergence to $p^{*}\left(\Psi |s,a,\lambda \right)$ vanishes for all a,λ and the two partial gradients are always nullvectors and therefore elements of the according nullspaces.

The free energy lemma can then be proven by showing that one of these three cases follows from the conditions of the lemma. However, no attempt was made in [1] to establish this. Instead, the given proof discusses the purported consequences of the existence of a suitable q(Ψ|λ). These will be discussed in Steps 5 and 6.

Even if the free energy lemma does not hold for systems obeying Equations (Equation 19) and (Equation 20), one might expect that the systems instead only satisfy the more general Equations (Equation 21) and (Equation 22) or the most general Equations (Equation 17) and (Equation 18). For these systems, the free energy lemma would require that there is a q(Ψ|λ) such that: (37)$$\begin{array}{cc} f_{a}\left(s,a,\lambda \right) & =\left(\left(\Gamma_{aa}+R_{aa}\right)\cdot \nabla_{a}+R_{a\lambda }\cdot \nabla_{\lambda }\right)F\left(s,a,\lambda \right), \end{array}$$(38)$$\begin{array}{cc} f_{\lambda }\left(s,a,\lambda \right) & =\left(R_{\lambda a}\cdot \nabla_{a}+\left(\Gamma_{\lambda \lambda }+R_{\lambda \lambda }\right)\cdot \nabla_{\lambda }\right)F\left(s,a,\lambda \right). \end{array}$$
or: (39)$$\begin{array}{cc} f_{a}\left(s,a,\lambda \right) & =\left(R_{as}\cdot \nabla_{s}+\left(\Gamma_{aa}+R_{aa}\right)\cdot \nabla_{a}+R_{a\lambda }\cdot \nabla_{\lambda }\right)F\left(s,a,\lambda \right), \end{array}$$(40)$$\begin{array}{cc} f_{\lambda }\left(s,a,\lambda \right) & =\left(R_{\lambda s}\cdot \nabla_{s}+R_{\lambda a}\cdot \nabla_{a}+\left(\Gamma_{\lambda \lambda }+R_{\lambda \lambda }\right)\cdot \nabla_{\lambda }\right)F\left(s,a,\lambda \right), \end{array}$$
hold, respectively. However, we find this not to be the case in general.

**Observation** **4.**
*Given a random dynamical system obeying Equation (Equation 1), ergodicity, Conditions 1 and 2, there need not exist a free energy expressed in terms of a variational density q(Ψ|λ) such that:*
**(i)** 
*Equations (Equation 29) and (Equation 30) hold if Equations (Equation 19) and (Equation 20) do;*
**(ii)** 
*Equations (Equation 37) and (Equation 38) hold if Equations (Equation 19) and (Equation 20) do not hold, but Equations (Equation 21) and (Equation 22) do;*
**(iii)** 
*Equations (Equation 39) and (Equation 40) hold if neither Equations (Equation 19) and (Equation 20) nor Equations (Equation 21) and (Equation 22) hold, but Equations (Equation 17) and (Equation 18) do.*



**Proof.** In Appendix C, we derive a set of conditions on the *R* and *U* matrices and on the putative variational density q(Ψ|λ), which follow from each of the pairs of equations in Cases (i–iii). We show that, in general, each pair leads to a contradiction, and in each case, we provide a counterexample that falls into the according system class. □

Before proceeding, we note that later works presented an alternative version of the free energy lemma, where the conditioning argument of q(Ψ|λ) was replaced by the most likely value of λ conditional on the (s,a) coordinates [2,3]. We here concern ourselves with the version apparent in [1], where q(Ψ|λ) is parametrised by the internal states themselves, but we briefly comment on the interpretation of the alternative approach in Step 7.

## 5. Vanishing Gradients

As mentioned in Step 4, the proof of the free energy lemma in [1] only discussed its consequences. The first proposed consequence is that expressing the vector field in terms of a free energy as in Equations (Equation 29) and (30) “requires” that the gradients with respect to *a* and λ of the KL divergence vanish, i.e., that Equations (Equation 35) and (36) hold.

We mentioned in Step 4 that the implication in the opposite direction holds. This can be seen from Equations (Equation 33) and (34). However, if the nullspace of ${\left(\Gamma +R\right)}_{aa}$ or ${\left(\Gamma +R\right)}_{\lambda \lambda }$ is non-trivial, then the gradient may be a non-zero element of this subspace and Equations (Equation 29) and (Equation 30) will still hold. In that case, the vanishing gradients would not be necessary for the free energy lemma.

The conditions under which a non-trivial nullspace exists were discussed in [7]. In short, the nullspace is guaranteed to be trivial in the special case where Γ is positive definite. Whether or not ergodic systems with a Markov blanket can ever admit a non-trivial nullspace, and hence divergences in Equations (Equation 31) and (Equation 32) with non-vanishing gradients, is not immediately clear. However, in order to establish the necessity of Equations (Equation 35) and (Equation 36), this remains to be proven.

## 6. Equality of Q(Ψ|λ) and $P^{*}\left(\Psi |s,a,\lambda \right)$

The proof of the free energy lemma in [1] also proposes that the vanishing of the gradients of the KL divergence, of the variational density q(Ψ|λ) from the conditional ergodic density $p^{*}\left(\Psi |s,a,\lambda \right)$, implies the equality of these densities. We mentioned in Equations (5) that the implication in the opposite direction holds. This can also be seen from Equations (Equation 33) and (Equation 34). Concerning the implication in the direction proposed by [1], let us now assume that for a given system of Equations (Equation 19) and (Equation 20) holds, a variational density q(Ψ|λ) does exist, and the gradients of the KL divergence of the variational and ergodic densities vanish, i.e., Equations (Equation 35) and (Equation 36) hold. Then, consider the argument by [1] in this direct quote (comments in square brackets by us):

“However, Equation (2.6) [Equations (Equation 19) and (Equation 20) above] requires the gradients of the divergence to be zero [Equations (Equation 35) and (Equation 36)], which means the divergence must be minimized with respect to internal states. This means that the variational and posterior densities must be equal:

$$\begin{array}{c} q\left(\psi |\lambda \right)=p^{\left[*\right]}\left(\psi |s,a,\lambda \right)\Rightarrow D_{\mathrm{KL}}=0\Rightarrow \left\{\begin{array}{cc} \; & \left(\Gamma +R\right)\cdot \nabla_{\lambda }D_{\mathrm{KL}}=0,\\ \; & \left(\Gamma +R\right)\cdot \nabla_{a}D_{\mathrm{KL}}=0. \end{array}\right. \end{array}$$

In other words, the flow of internal and active states minimizes free energy, rendering the variational density equivalent to the posterior density over external states.”

The first problem in the above quote is that the minimization of the divergence does not follow from the vanishing gradients. On the contrary, since Equations (Equation 35) and (36) must hold for all (s,a,λ), the KL divergence:$$D_{\mathrm{KL}}\left[q\left(\Psi |\lambda \right)|\right|p^{*}\left(\Psi |s,a,\lambda \right)]$$
cannot depend on (λ,a); it therefore has no extremum (and thus no minimum) with respect to either of these coordinates.

The second problem pertains to the identification of the two distributions at a minimum. In general, if we try to find the minimum of a KL divergence between a given probability density $p_{1}\left(Y\right)$ and a family of densities $p_{2}\left(Y|\theta \right)$ parameterized by θ, then the lowest possible value of zero is achieved only if there is a parameter $\theta_{1}$ such that $p_{2}\left(Y|\theta_{1}\right)=p_{1}\left(Y\right)$. If there is no such $\theta_{1}$, then the minimum value will be larger than zero. Therefore, even if the divergence were minimized, it would not need to be zero. More generally, the divergence K(s) need not be zero for any value of *s*.

There is therefore no satisfactory reason given why the variational density q(Ψ|λ) and the posterior density $p^{*}\left(\Psi |s,a,\lambda \right)$ should be equal or have low KL divergence. In fact, they need not be (Note that, since any q(Ψ|λ) that does not depend on (s,a) is an element of the set of those that do, Observation 5 remains true for the case where we allow this dependence. In that case, the free energy lemma holds because we can set $q\left(\Psi |s,a,\lambda \right):=p^{*}\left(\Psi |s,a,\lambda \right)$, and thus, a *q* exists for which the densities are actually equal. However, the claim here is that for every *q* that obeys the conditions in Observation 5, we must have equality.).

**Observation** **5.**
*Given a random dynamical system obeying Equation (Equation 1), ergodicity, Conditions 1 and 2. Then if, additionally,*
**(i)** 
*Equations (Equation 19) and (Equation 20) hold and the free energy lemma holds, i.e., there exists a probability density q(Ψ|λ) such that Equations (Equation 29) and (Equation 30) hold, or*
**(ii)** 
*Equations (Equation 21) and (Equation 22) hold and there exists q(Ψ|λ) such that Equations (Equation 37) and (Equation 38) hold, or*
**(iii)** 
*Equations (Equation 17) and (Equation 18) hold and there exists q(Ψ|λ) such that Equations (Equation 39) and (Equation 40) hold,*

*then there is no c≥0 for which it can be guaranteed that:*
(41)$$D_{KL}\left[q\left(\Psi |\lambda \right)|\right|p^{*}\left(\Psi |s,a,\lambda \right)]<c.$$
*In particular, it does not follow from these conditions that:*
(42)$$\begin{array}{c} q\left(\Psi |\lambda \right)=p^{*}\left(\Psi |s,a,\lambda \right). \end{array}$$


**Proof.** By example, see Appendix D. To show that the implication does not generally hold for a given system and densities q(Ψ|λ) that obey Equations (Equation 19), (Equation 20), (Equation 29), and (Equation 30), Equations (Equation 21), (Equation 22), (Equation 37), and (Equation 38), or Equations (Equation 17), (Equation 18), (Equation 39), and (Equation 40), we only have to consider a system that obeys all three pairs of equations, Equations (Equation 19) and (Equation 20), Equations (Equation 21) and (Equation 22), and Equations (Equation 21) and (Equation 22), and for which a suitable q(Ψ|λ) exist. For this system, we then need to show that the q(Ψ|λ) that obey Equations (Equation 29) and (30) are not necessarily equal (or similar) to $p^{*}\left(\Psi |s,a,\lambda \right)$.We use a variant of the model used in Appendix B as such a counterexample. This system obeys all three of Equations (Equation 19) and (Equation 20), Equations (Equation 21) and (Equation 22), and Equations (Equation 21) and (Equation 22), and the nullspace of the associated Γ+R is trivial. We identify a set of possible q(Ψ|λ) satisfying Equations (Equation 29) and (30), which implies that the gradients of the KL divergence between those q(Ψ|λ) and $p^{*}\left(\Psi |s,a,\lambda \right)$ vanish, i.e., Equations (Equation 35) and (36) hold. We then demonstrate that for the q(Ψ|λ) in this set, the value of the KL divergence to $p^{*}\left(\Psi |s,a,\lambda \right)$ can be arbitrarily large. □

## 7. Interpretation

Finally, we turn our attention to the interpretation in terms of Bayesian inference, i.e., Step 7. We again quote directly from [1]:

Because (by Gibbs inequality) this divergence [D_KL_$\left[q\left(\psi |\lambda \right)||p^{*}\left(\psi |s,a,\lambda \right)\right]$] cannot be less than zero, the internal flow will appear to have minimized the divergence between the variational and posterior density. In other words, the internal states will appear to have solved the problem of Bayesian inference by encoding posterior beliefs about hidden (external) states, under a generative model provided by the Gibbs energy.

We showed that, in general, there is no suitable variational density that is only parameterized by the internal coordinate λ. We then showed that, even if there is a suitable variational density (including those parameterized by all of (s,a,λ)), it can be arbitrarily different from the posterior density. Since the arguments for the internal flow appearing to minimize the divergence between variational and posterior density are therefore incorrect, there is no reason why the internal states should appear to have solved the problem of Bayesian inference.

As mentioned in Step 4, some newer works (e.g., [2,3]) formulated a different free energy principle, where the variational density of beliefs is parametrised not by the internal coordinates λ, but by $\bar{\lambda }\left(s,a\right)={arg max}_{\lambda }p^{*}\left(\lambda |s,a\right)$, the most likely value of the internal coordinates given the sensory and active ones. In this case, Observations 4 and 5 do not apply. However, the new parameters $\bar{\lambda }\left(s,a\right)$ are strictly a function of the sensory and active coordinates. This means we have a Markov chain (with capitalisations indicating random variables associated with the corresponding lower case coordinates (or functions of coordinates)) $\Lambda \to \left(S,A\right)\to \bar{\Lambda }$ and, by the data processing inequality [14], the mutual information between the both sensory and active coordinates and the belief parameter $\bar{\lambda }$ upper bounds that are between the internal coordinates and the belief parameter. It is therefore not clear to what extent the internal coordinates λ, rather than the active and sensory coordinates (s,a) themselves, can be said to be encoding beliefs about the external coordinates. Note also that, on any given trajectory, unless the distribution $p^{*}\left(\lambda |s,a\right)$ is sufficiently peaked and unimodal, the internal coordinates are not guaranteed to spend most of their time close to their most likely conditional value, and (by definition if Condition 2 holds) they will not be better predictors of the external coordinates than those in the Markov blanket.

Generally, $\lambda \neq \bar{\lambda }$, and $\bar{\lambda }$ is the solution to an optimization problem that is assumed to be solved in these later works. Using this optimized variable to parametrise beliefs is therefore a considerable departure from [1]. Contrary to the impression created by the way it was referenced in [2,3], the older theory in [1] should be clearly distinguished from the newer ones in these more recent papers.

## 8. Consequences for Friston, K. et al. 2014

Reference [4] argued for the same interpretation as [1], but there were some differences in the argument.

The differences were the following:In [4], Equation (Equation 1) was formulated for “generalized states”, which we refer to here as generalized coordinates. This means that the variable *x* is replaced by a multidimensional variable denoted $\tilde{x}=\left(x,x^{\prime },x^{\prime \prime },...\right)$.The Markov blanket structure was not explicitly defined via Equation (Equation 2). Formally, it was introduced directly (see [4] Equation (Equation 10)) in a less general form corresponding to Equations (Equation 19) and (20) (at the same time, [1] is referenced in connection to the Markov blanket so there seems to be no intention to replace the original definition with the stronger one). Therefore, our observations concerning Steps 2 to 4 are not directly relevant to this paper.The internal coordinate λ was renamed to *r*, and the role of matrix *R* was played by the matrix −Q.The proof of the free energy lemma given in [4] was different. It (implicitly) suggested setting the variational density equal to the ergodic conditional posterior.The proof of the free energy lemma no longer contained the proposition that the gradient of the KL divergence of the variational density and the ergodic conditional density vanish, i.e., Step 5.The proof also no longer contained the claim that the vanishing gradients of the KL divergence of the variational density and the ergodic conditional density imply the equality of those densities, i.e., Step 6 was not present.

The interpretation in terms of Bayesian inference was unchanged and still relied on the equality of the variational and the ergodic conditional density.

Since there were no explicit generalized coordinate versions of Steps 2, 3, 5 and 6 in [4], we do not discuss those steps here. We only disprove the free energy lemma and the claim that when the free energy lemma holds, the variational and ergodic conditional density become equal. For this, we present a way to translate the counterexamples used in Observations 4 and 5 into counterexamples in generalized coordinates. The interpretation in terms of Bayesian inference given in [4] is therefore equally as unjustified as the one in [1].

For completeness, we first state the generalized coordinate versions of the stochastic differential Equation (Equation 1):(43)$$\begin{array}{c} \dot{\tilde{x}}=f\left(\tilde{x}\right)+\tilde{\omega }, \end{array}$$
the less general version of the Markov blanket structure Equation (Equation 2):(44)$$\begin{array}{c} \begin{array}{cc} f_{\tilde{\psi }}\left(\tilde{\psi },\tilde{s},\tilde{a}\right) & ={\left(\Gamma -Q\right)}_{\tilde{\psi }\tilde{\psi }}\nabla_{\tilde{\psi }}\ln p^{*}\left(\tilde{\psi },\tilde{s},\tilde{a},\tilde{r}\right)\\ f_{\tilde{s}}\left(\tilde{\psi },\tilde{s},\tilde{a}\right) & ={\left(\Gamma -Q\right)}_{\tilde{s}\tilde{s}}\nabla_{\tilde{s}}\ln p^{*}\left(\tilde{\psi },\tilde{s},\tilde{a},\tilde{r}\right)\\ f_{\tilde{a}}\left(\tilde{s},\tilde{a},\tilde{r}\right) & ={\left(\Gamma -Q\right)}_{\tilde{a}\tilde{a}}\nabla_{\tilde{a}}\ln p^{*}\left(\tilde{\psi },\tilde{s},\tilde{a},\tilde{r}\right)\\ f_{\tilde{r}}\left(\tilde{s},\tilde{a},\tilde{r}\right) & ={\left(\Gamma -Q\right)}_{\tilde{r}\tilde{r}}\nabla_{\tilde{r}}\ln p^{*}\left(\tilde{\psi },\tilde{s},\tilde{a},\tilde{r}\right), \end{array} \end{array}$$
the expression of the $\tilde{a}$ and $\tilde{r}$ components of the vector field in terms of the marginalised ergodic density Equations (Equation 19) and (20): (45)$$\begin{array}{cc} f_{\tilde{a}}\left(\tilde{s},\tilde{a},\tilde{r}\right) & ={\left(\Gamma -Q\right)}_{\tilde{a}\tilde{a}}\cdot \nabla_{\tilde{a}}\ln p^{*}\left(\tilde{s},\tilde{a},\tilde{r}\right), \end{array}$$(46)$$\begin{array}{cc} f_{\tilde{r}}\left(\tilde{s},\tilde{a},\tilde{r}\right) & ={\left(\Gamma -Q\right)}_{\tilde{r}\tilde{r}}\cdot \nabla_{\tilde{r}}\ln p^{*}\left(\tilde{s},\tilde{a},\tilde{r}\right), \end{array}$$
and in terms of free energy Equations (Equation 29) and (30): (47)$$\begin{array}{cc} f_{\tilde{a}}\left(\tilde{s},\tilde{a},\tilde{r}\right) & ={\left(Q-\Gamma \right)}_{\tilde{a}\tilde{a}}\cdot \nabla_{\tilde{a}}F\left(\tilde{s},\tilde{a},\tilde{r}\right), \end{array}$$(48)$$\begin{array}{cc} f_{\tilde{r}}\left(\tilde{s},\tilde{a},\tilde{r}\right) & ={\left(Q-\Gamma \right)}_{\tilde{r}\tilde{r}}\cdot \nabla_{\tilde{r}}F\left(\tilde{s},\tilde{a},\tilde{r}\right). \end{array}$$
The free energy lemma then requires that there exists $q(\tilde{\Psi }|\tilde{r})$ such that the KL divergence between $p^{*}\left(\tilde{\Psi }|\tilde{s},\tilde{a},\tilde{r}\right)$ vanishes. Without going into further details of the difference between the proof in [4] and that in [1], we can prove the former wrong by translating the counterexample used for the latter into generalised coordinates.

**Observation** **6.**
*There is a general way to translate a system in ordinary coordinates into a system of generalised coordinates that corresponds to an infinite number of independent copies of the original system. This means all properties of the original system (e.g., linearity, ergodicity, the Gaussian and Markovian property of the noise, Conditions 1 and 2, the properties of Γ,R,U) are preserved during this translation.*


**Proof.** By construction, see Appendix E. □

This implies that the counterexamples used in proving Observations 4 and 5 directly translate to the setting of the generalised coordinates. The free energy lemma is therefore also wrong for generalised coordinates, and the variational density $q(\tilde{\Psi }|\tilde{r})$ is not “ensured” [4] to be equal to the conditional ergodic density $p^{*}\left(\tilde{\Psi }|\tilde{s},\tilde{a},\tilde{r}\right)$.

## 9. Conclusions

We find that the two different Markov blanket conditions proposed in [1,2,3] are independent of each other. We then show that under both of those Markov blanket conditions, among the six steps contained in the argument in [1], three do not hold independently of each other. We also show that fixing the second of those steps (Step 3) does not provide a valid alternative. The line of reasoning of [1] therefore does not support its claim that the internal coordinates of a Markov blanket “appear to have solved the problem of Bayesian inference by encoding posterior beliefs about hidden (external) [coordinates], …”. We also show that using generalised coordinates as in [4] does not remedy the situation. Additionally, we identify a technical error in [2] and an interpretational issue resulting from possibly too strong assumptions (both Conditions 1 and 3) in [3]. We also highlight that the latter publications both argued that it is the most likely internal coordinates given sensory and active coordinates that encode posterior beliefs about external states instead of the internal coordinates themselves. The resulting free energy principle and lemma are therefore a different proposal. This is not subject to our technical critique.

## Figures and Tables

**Figure 1 entropy-23-00293-f001:**
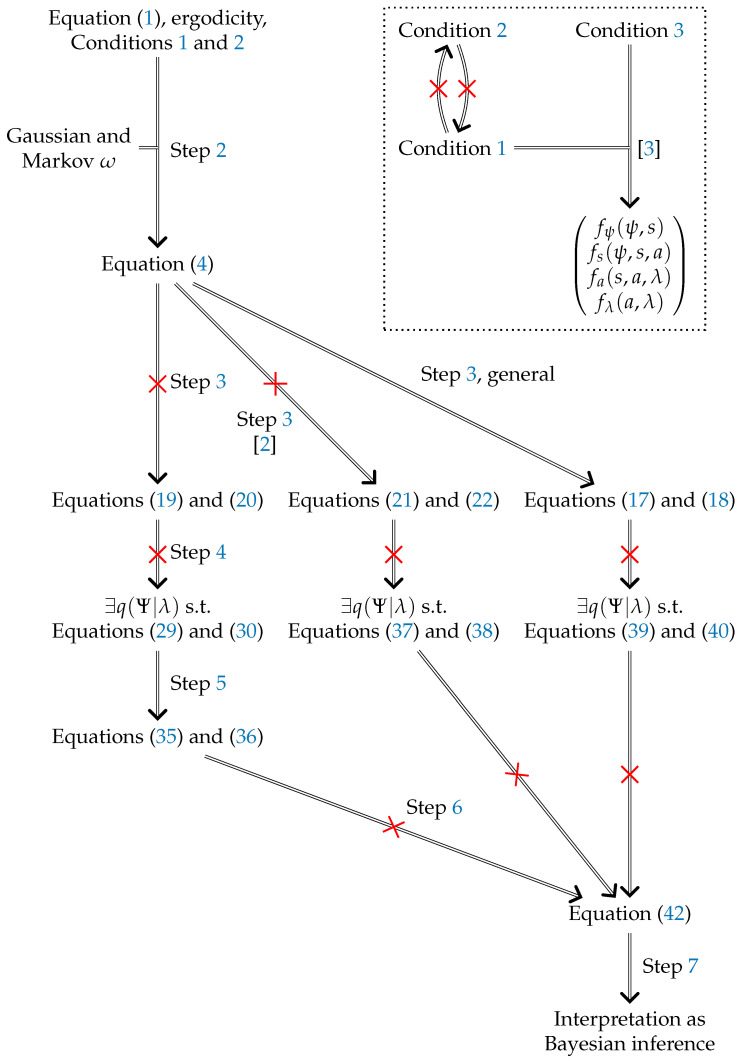
Argument visualization. Numbers labelling edges indicate corresponding steps in this paper. Struck out edges indicate implications that we prove incorrect. The main argument in [1] takes the left path. The box in the top right indicates the relations between Conditions 1 to 3 and their role in [3]. Merged edges indicate a logical AND combination of the parent nodes.

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
