# Peer review of "A Technical Critique of Some Parts of the Free Energy Principle"

_entropy, 2021, doi:10.3390/e23030293_

Round 1

Reviewer 1 Report

Authors has made an effort to consider reviewer's suggestions. Now it is clear that there is no critique to the general framework of the Commented work, though authors still harbor a mild suspicion that   " In general, it is not clear that Eq. (4) holds in the non-Markovian case, since the standard derivations in Ao [6], Kwon et al. [7] and related works rely on delta-correlated noise."   It may be pointed out the general framework should be valid for non-Markovian situations:  the present reviewer has been advised by mathematicians that for linear dynamics rigorous demonstration has been carrying out for non-Markovian case.

On the technical side, it is clear that the authors observed a few weak points. While such weak points, if exist, could be addressed by the commented authors, for such important topic the manuscript has a value of publication.  

Reviewer 2 Report

Please see the attached pdf file.

Reviewer 3 Report

The series works by Karl Friston et al. present a heuristic argument for one of the most intriguing issues -- how life emerged. I appreciate the argument which is a brave attempt and it is certainly not a rigorous mathematical proof yet. Technical details remain to be established. The work by Martin Biehl et al. pointed out a few of such problems. I think the technical questions raised by Martin Biehl et al. are addressable. For example, I agree that some equations proposed by Friston et al. may not be comprehensive enough so far, therefore it leaves space for counterexamples, e.g. "all of the off-diagonal terms are implicitly assumed to vanish" for Eq. 19 and 20. But I think they are generalizable to include off-diagonal terms.  

Overall I would suggest the authors to tone down their statements. I believe the free energy approach can generally be used in explaining fundamental questions such as biological self-organization. I would like to see how Karl Friston et al. response to this technique critique. 

Author Response

Please see PDF.

This manuscript is a resubmission of an earlier submission. The following is a list of the peer review reports and author responses from that submission.

Round 1

Reviewer 1 Report

I believe that the authors have raised some important issues on a well known free energy approach to biological systems especially applied to neural network dynamics. The questions they raised seem reasonable and require further careful studies. For example, whether the free energy can be rigoriously defined for a nonequilibirum system is still under debate in the field. It is good to bring this up and give appropriate attentions. This paper is publishable and expected to contribute to the better understanding of the dynamics for the biological systems. 

Reviewer 2 Report

The manuscript raised concerns to a promising approach to one of most important scientific problems. In particular, authors asserted to find  three formal errors in the promising approach:     

i) The rewriting of the equations of motion of systems with Markov blankets which turns out not to be generally correct. We prove the non-equivalence with a counterexample that exhibits a Markov blanket but does not satisfy the rewritten equations. Our counterexample also invalidates the corresponding (but more general) rewritten equations in the more recent Friston.

ii) The Free Energy Lemma itself, which we prove, by counterexample, to be wrong in general.

iii) The Free Energy Lemma, when it does hold, implies equality of variational density and ergodic conditional density.   The interpretation in terms of Bayesian inference hinges on this point, and we hence conclude that it is unjustified.  

Nevertheless, there are many obvious errors in the manuscript which prevents a clear assessment of manuscript. The authors should consider them carefully before resubmission for publication.  

1)  Their Eq.(1) should be not called Langevin equation, which has a specific meaning in physics that the potential function in phase space has been known and (generalized) detailed balance is satisfied, that is, R = 0, as R used in Eq.(4) of this manuscript.  

A more neutral and suitable term, stochastic differential equation, should be used: To start with, only f and \omega are know. 

Potential function is an emerged quantity of such dynamical process. 

With this line of reasoning, Eq.(2) is of primary importance.   

With Eq.(1) and (2), potential energy always exists, hence, "free energy" can be generally defined. The rest is of secondary importance.  

2) Indeed, Eq.(4),(5),(6) are exact only for linear dynamics, their Ref.[6], and are approximated correct for nonlinear situations as the lowest order of so-called gradient approximation.   

In nonlinear situations, Eq.(4) is still valid, it is Eq.(5) to be modified. This was expressed clearly in their Ref.[9].

Hence, the authors need to explain their assertion, " for general nonlinear models, …  Eq. (4) must be modified". 

3) For their statement, "The exact conditions under which these matrices can be chosen to be constant are not known to us", the equation to determine those quantities exactly had written down in their Ref.[9] . 

The authors have to explain for what reasons that they believe conditions were not known.  

There has been a series of efforts to explore various situations of that condition. A recent summary on the exact condition may be found  Yuan et al., Frontiers of Physics 12 (2017) 120201, SDE decomposition and A-type stochastic interpretation in nonequilibrium processes.        

4)  It has been known that one should not mix up conditions on potential energy U and on force matrix M. Examples have been known in literature, for example,  Yuan et al., Chinese Physics B 2014, 23 (1): 010505: Lyapunov Function as Potential Function: A Dynamical Equivalence.  

5) In Appendix A, it would be better to explicitly write down R .

6) In Appendix E, their assertion,

" In general, the Kramers-Moyal coefficients will not vanish beyond second order, meaning that an approach for which there is an equivalent Fokker-Planck equation, and hence Eq. (4), is not valid [14]. " 

missed the key point of the existence of "potential function".  

The structure of Eq.(4), the "diffusion matrix "Γ" (detailed balance), cyclic motion ( "R" ), "potential" energy, is the same for a Markov process,  see for example,  Ao et al.,  Chinese Physics Letters, 2013, 30 (7) ,070201: Dynamical Decomposition of Markov Processes without Detailed Balance.  This implies that "free energy" can always be defined.

Reviewer 3 Report

Please see the attached pdf file.
